# The Effects of a New Integrated and Multidisciplinary Cognitive Rehabilitation Program Based on Mindfulness and Reminiscence Therapy in Patients with Parkinson’s Disease and Mild Cognitive Impairment: A Pilot Study

**DOI:** 10.3390/brainsci13020201

**Published:** 2023-01-25

**Authors:** Maria Rita Reitano, Matteo Guidetti, Natale Vincenzo Maiorana, Angelica De Sandi, Fabrizio Carusi, Chiara Rosci, Fabiana Ruggiero, Barbara Poletti, Nicola Ticozzi, Francesca Mameli, Sergio Barbieri, Vincenzo Silani, Alberto Priori, Roberta Ferrucci

**Affiliations:** 1ASST Santi Paolo e Carlo, San Paolo University Hospital, 20142 Milan, Italy; 2Aldo Ravelli Center for Neurotechnology and Experimental Brain Therapeutics, Department of Health Sciences, University of Milan, 20142 Milan, Italy; 3Department of Electronics, Information and Bioengineering, Politecnico di Milano, 20133 Milan, Italy; 4Foundation IRCCS Ca’ Granda Ospedale Maggiore Policlinico, 20122 Milan, Italy; 5Department of Neurology and Laboratory of Neuroscience, IRCCS Istituto Auxologico Italiano, 20149 Milan, Italy; 6Department of Pathophysiology and Transplantation, Dino Ferrari Center, University of Milan, 20122 Milan, Italy

**Keywords:** mindfulness, reminiscence and life review, verbal long-term memory, Parkinson’s disease, mild cognitive impairment

## Abstract

*Background*: Mindfulness trainings have shown promising results as treatment for behavioural symptoms in several pathologies. In addition, mindfulness protocols induced an improvement in memory and attention. Therefore, mindfulness could be an effective intervention for patients affected by Parkinson’s disease (PD) and mild cognitive impairment (MCI), who are characterized by both behavioural and cognitive dysfunctions. *Methods*: We assessed differences in Montreal Cognitive Assessment (MoCA) scores and in Beck Depression Inventory II (BDI-II) scores in patients affected by PD and MCI enrolled in two different rehabilitation programs (an experimental vs. an usual structured program for cognitive rehabilitation). Participants in the experimental group (MILC-tr) underwent innovative rehabilitation program involving mindfulness and reminiscence activities. Assessments were performed before (T0) and at the end of the rehabilitation program (T1). *Results*: Friedman test showed a significant improvement between timepoints in MoCA global score (x^2^ = 4.000, *p* = 0.046), MoCA memory sub-scale score (x^2^ = 4.571, *p* = 0.033), and BDI-II cognitive and affective factors (x^2^ = 4.000, *p* = 0.046) only for patients in MILC-tr group. Mann–Whitney test showed a significant difference between group comparing differences in Δ scores between T0 and T1 in the MoCA memory sub-scale score (U = 190.50, *p* = 0.035). *Conclusions*: Mindfulness-based rehabilitation programs could be effective in patients affected by PD and MCI.

## 1. Introduction

Mild cognitive impairment (MCI) is a condition characterized by a cognitive impairment greater than that due to normal aging, but not severe enough for a diagnosis of dementia [1,2] nor for a significant impact on the activities of daily living [3]. Although the true values are difficult to define [4], the literature suggests that prevalence of MCI in elderly people (>65 years old) would be 3–22% [2,5,6], depending on the demographics of the population studied. Although in the past MCI was considered simply a “precursor” to or a “paucisymptomatic” phase of Alzheimer’s disease (AD) [7], not all cases of MCI anticipate dementia diagnosis, and not all are progressive [3]. Often, MCI is associated with depressive symptoms that negatively impact the quality of life [1], and with other medical disorders, like traumatic brain injury, cerebrovascular accident, and Parkinson’s disease (PD) [8]. PD is a common neurodegenerative disease where the progressive degeneration of the dopaminergic neurons in the substantia nigra leads to decrease in the dopamine secretion of the striatum [9,10]. Clinically, the cardinal symptoms of PD include motor- and non-motor-related features. Motor symptoms include bradykinesia, rigidity, static tremor, and postural and gait disorder [11]; non-motor symptoms include psychiatric (e.g., psychosis, apathy, behavioural changes) [12] and cognitive disorders [13]. Depression is a very frequent symptom in PD, and occurs in 40–50% of patients [14]. In addition, it was estimated that 15% to 40% of patients with PD meet MCI criteria at diagnosis [15,16], and that ~50% of them develops MCI during the disease [17]. Currently, no resolutive or disease-modifying therapy is available for MCI and PD [18], and cognitive rehabilitation plays a pivotal role in maintaining residual resources and patients’ autonomy in daily life activities, and in promoting psychological wellbeing [19,20]. Several non-pharmacological approaches have been proposed to improve MCI and PD patients’ quality of life. Among these, promising results were disclosed by mindfulness training [21,22,23,24]. 

Mindfulness is a practice that conceptually derives from the millennia old history and broader concept of mediation [25]. It specifically aims to increase the personal awareness of the moment, fully focusing on internal and external experiences as they occur in the actual present, with the suspension of personal judgments and openness to these current experiences [26,27,28,29]. Stripped of all religious aspects, mindfulness-based interventions (e.g., mindfulness-based stress reduction trainings—MBSR, or mindfulness-based cognitive therapy—MBCT) have been shown to reduced symptomatology in anxiety disorders [30], depression, and post-traumatic stress disorder [31]; to reduce stress levels in healthy subjects [32]; and to increase emotional stability, mood, and emphatic abilities [33]. A study by Hölzel et al. (2011) [34] found that participants who engaged in an eight-week mindfulness meditation program had increased grey matter density in the hippocampus, a brain region involved in learning and memory, as well as in the temporo-parietal junction, a brain region involved in perspective-taking and empathy. In addition, significant changes in the cerebellum, a brain structure involved in depressive disorders [35,36,37]. Similarly, Luders et al. (2009) [38] found that long-term meditation practitioners had increased grey matter volume in the prefrontal cortex, a brain region involved in attention and executive function, as well as in the insula, a brain region involved in enteroception and empathy. This research suggests that mindfulness meditation programs can have a positive effect on brain structures involved in emotional regulation and cognition [39]. Promising results regarding emotional regulation were found in different healthy and in pathological populations [25]. A positive effect of meditation training on attention, memory, and self-reported improvements of speech have been reported in PD patients [21] and healthy subjects [40], with mindfulness-based interventions slowing down cognitive decline in patients with MCI and AD [22]. For example, Yu et al. (2021) [24] studied the cognitive and neurophysiological (cortical thickness) effects of a nine-month lasting mindfulness-based intervention on MCI subjects, as compared to an active control treatment (health education program). Patients assigned to the mindfulness group had greater performances in working memory span and divided attention after treatment, and increased cortical thickness in the right frontal lobe [24]. Several results suggest that mindfulness-induced psychological and behavioural effects correlate with modifications in brain function and structure [25], both in healthy subjects [41,42], as well as in people with AD [43,44] and PD [23]. For example, a neurobiological study [23] found that in patients with PD, mindfulness-based intervention increased grey matter density in the right and left hippocampus, part of the right amygdala, and left and right caudate nucleus, consistently with improved emotional regulation [25]. However, results on PD and MCI subjects are still limited, and a standardized mindfulness-based protocol for cognitive rehabilitation is lacking.

In this study, we evaluate the effects of a new mindfulness-based program (Mindfulness Integrated Life-review and Cognitive Training—MILC-tr) on cognitive functions and mood in patients with PD and MCI, compared to conventional Cognitive Rehabilitation Training (CR-tr). We hypothesized that MILC-tr could lead to a major improvement in patients’ cognitive abilities and mood.

## 2. Materials and Methods

### 2.1. Participants

Thirty-three patients (age: mean ± SD: 70.5 ± 11, 15 females) with a clinical diagnosis of idiopathic PD (n = 11) or MCI (n = 22) were recruited from January 2018 to June 2022. Patients were included in this study according to the following criteria: (1) age between 40 and 85 years; (2) PD or MCI diagnosis; (3) MoCA score ≥ 12; (4) years of education ≥ 5 (i.e., level of education superior or equal to primary school); (6) ability to express their consent to participate in the study. The following exclusion criteria were applied: (1) MoCA score < 12; (2) presence of neuropsychiatric comorbidities; (3) education years < 5; (4) age < 40 years or >86 years; (5) visual or auditory deficits. All subjects gave written informed consents before the participation, and the study was approved by the Institutional Review Board and conducted in compliance with the Declaration of Helsinki.

### 2.2. Rehabilitation Protocols

Experimental group underwent a new neurocognitive rehabilitation protocol (MILC-tr), which considers different aspects of residual cognitive functions, with the aim to enhance or compensate them. The protocol is structured in three different phases: (1) Formal Mindfulness practice, which is a short adaptation of the MBSR program [28] that includes the ‘body-scan’ procedure—15 min; (2) Life Review and Reminiscence Therapy/Life story book [45] and the Autobiographical memory (ABM) interview [46] and the Autobiographical Memory of Crovitz-Schiffman [47] (used with word-cued technique to train long-term autobiographical memory—15 min); (3) process-based cognitive exercises and strategic/metacognitive training (15 min). In particular, Life Review and Reminiscence Therapy was developed by Butler in 1963 [48] as a dementia treatment by using a “life-review approach”, meaning that various memory triggers such as household objects, past photographs, and music may be used to recall autobiographical events able to enrich life experiences of elderly people. It became widely applied in 80s [49], and it is now an established and recommended therapy for cognitive deterioration and depressive symptoms, because it specifically supports people in affirming residual cognitive abilities (especially memory ones), and in enhancing self-esteem and interpersonal skills [50,51,52].

Control group underwent a structured program for cognitive rehabilitation of patients with neurodegenerative diseases (CR-tr). It consists of two phases: (1) process-based cognitive exercises (repeated and gradual practice of tests involving target skills like memory, attention, executive functions—30 min); (2) strategic and metacognitive training (strategic use of target skills and metacognitive training oriented to modify dysfunctional beliefs about one’s cognitive functioning—10 min).

### 2.3. Study Design 

In this randomized, controlled, exploratory study, each participant was randomly assigned to either experimental group (MILC-tr; n = 19) or control group (CR-tr; n = 14), using a random number generator. In both cases, patients underwent treatment sessions lasting 50 min, 1 day per week for 8 weeks. Cognitive (MoCA) and psychological (BDI-II) assessments were conducted before (T0) and at the end (T1) of the treatment.

### 2.4. Cognitive and Psychological Assessment

Each patient was assessed before (T0) and at the end (T1) of the treatment. To assess cognitive functions, the Montreal Cognitive Assessment (MoCA) was administered [53,54]. MoCA is a rapid screening battery, including several tasks with visuospatial material, together with tasks and procedures specifically assessing frontal functions, attention, and long-term memory. It evaluates 6 cognitive domains, i.e., memory (Long Term Verbal Memory—score range 0–5), visuospatial abilities (copying cube, clock drawing task—score range 0–4), executive functions (brief version of the Trail making test, phonemic fluency task, abstract reasoning—score range 0–4), attention, concentration, working memory (digit span forward, digit span backward, sustained attention task, series of 7s—score range 0–6), language (naming task with low-familiarity animals, repetition of two sentences, phonemic fluency task—score range 0–6), spatiotemporal orientation (score range 0–6), and allows to characterize cognitive profile of the patient [54]. The total score is 30, with values ≤ 26 indicative of cognitive impairment [54]. Similar to previous literature [55,56], each subdomain may be considered singularly.

To assess depression symptoms, the Beck Depression Inventory II (BDI-II) was administered [57]. BDI is a self-assessment psychological questionnaire to detect the severity of depression in adults and teenager (≥13 years old). If patients present problems of reading and concentration, it can be read by an interviewer. The questionnaire consists of 21 items, and each is scored on a scale of 0–3, with higher total score meaning more severe condition (0–13 indicates minimal depression; 14–19 indicates mild depression; 20–28 indicates moderate depression; 29–63 indicates severe depression) [58]. BDI-II assumes that depression is characterized by two components: the mental one and the somatic one. Therefore, the items might be categorized in cognitive-affective factors (Sadness, Pessimism, Failure, Guilt, Feeling punishment, Self-esteem, Self-criticism, Suicide, Feeling of worthlessness) and somato-affective factors (Crying, Loss of pleasure, Agitation, Loss of interest, Loss of energy, Indecision, Sleep, Irritability, Appetite, Concentration, Fatigue, Sex) [59]. The two factors can be considered singularly [60,61].

### 2.5. Statistical Analysis

Normality of the distribution was assessed on MoCA and BDI-II scores using the Shapiro–Wilk test of normality. Due to the test result (*p* < 0.05), nonparametric statistics were used for data analysis. For each variable, the Friedman test was applied to compare the score changes between T0 and T1 within subjects in the two groups. The Mann–Whitney U test was performed to assess the difference between groups at each timepoint, as for the formula: Δ score = T1 score − T0 score, for both MoCA and BDI-II overall scores and for their respective sub-scale scores. Results were considered statistically significant for *p* < 0.05. All data were analysed using JASP software (Version 0.16.3).

## 3. Results

The sociodemographic characteristics of the sample are displayed in Table 1. Cognitive and psychological scores are reported in Table 2 and Table 3.

Male sex was the most representative in our sample (54.6%). Patients in MILC-tr group were younger (mean ± SD: 68.8 ± 10.6 y.o.) than CR-tr group (mean ± SD: 72.7 ± 11.5) and had higher education (MILC-tr group: 12.31 ± 3.4 y.o.; CR-tr group: 12.31 ± 3.4). One third of the study sample received diagnosis of PD (33.3%, n = 11), while the most were diagnosed with MCI (66.6%, n = 22). Friedman test showed a significant improvement between timepoints in MoCA global score (χ^2^ = 4.000, *p* = 0.046) and MoCA memory sub-scale score (χ^2^ = 4.571, *p* = 0.033), only for patients in MILC-tr group (see Figure 1). All other analysis between pre- and post- treatment assessments did not show statistical significance (all tests *p* > 0.05). Mann–Whitney U test disclosed a significant difference between groups only in Δ MoCA memory sub-scale score, which was significantly higher in experimental group compared to control (U = 190.50, *p* = 0.035) (see Figure 2).

## 4. Discussion

Mindfulness-based interventions have been shown to enhance residual cognitive abilities and slow down cognitive decline in patients with PD [21,25] and MCI [22,24]. In this exploratory study, we assess the effects of a new mindfulness-based protocol of cognitive rehabilitation on cognitive functions and depression in people with PD and MCI and compare the results to a conventional Cognitive Rehabilitation Training. Our results revealed an improvement in overall cognitive functions and in long term verbal memory only in the experimental group, These results are in line with recent studies that showed that mindfulness leads to significant changes in cognitive functions such as attention, memory, and executive functions, in healthy and pathological subjects [62,63,64,65,66]. After mindfulness-based training, a significant improvement in working memory [67,68,69] and an increased ability to voluntarily focus on the personal awareness of the moment, improving selective, executive, and sustained attention [70,71,72,73] was found. In MCI patients, mindfulness-based interventions might slow down cognitive decline [22], while in PD patients it improved attention, memory, and self-reported assessment of speech [21]. Our experimental protocol might have improved such cognitive domains by the stimulation of senses and the structured recall of autobiographical memories. Indeed, Janssen et al. (2015) [74] found that verbal and visuo-spatial memory are correlated with autobiographical memory, according to the basic systems approach [75,76]. This approach assumes that autobiographical memory is supported by other cognitive systems, as a sort of multimodal entity that can involve sight, hearing, smell, taste, touch, language and can vary greatly in spatial, temporal, emotional, and narrative content. Therefore, when the senses are stimulated in rehabilitation programs such as MILC-tr, it is possible to hypothesize a potential beneficial effect on memory. In addition, neuroimaging studies have described an actual effect of mindfulness techniques in modifying brain function and structure in several population (e.g., people with Alzheimer’s disease [43,44] and people with PD [23]) which is consistent with clinical psycho-cognitive improvements [25]. Indeed, after mindfulness-based training, the prefrontal and cingulate cortex, the insula, and the hippocampus increased their activity, while the amygdala decreased it [25], consistently with enhanced meta-awareness and reappraisal, body awareness, memory processes, and emotional regulation [77,78,79]. We have found significant changes in depressive symptoms in BDI-II cognitive affective factors, as previously suggested [80]. This finding might be in line with the conceptual focus of our MILC-tr intervention, since it emphasizes the importance of psychological well-being through present moment awareness and acceptance, self-exploration, and control over their situations. Our experimental treatment might have encouraged a new way of thinking about disability, motivating a sense of acceptance. On the other side, no effects were detected in BDI-II total score. This finding is in partial agreement with those reported by Pickut et al. 2015 [81] who, however, focused only on total BDI-II score without considering the BDI-II subscales. It should be noted that mindfulness practice eases the development of several attitudes like acceptance, openness, suspension of judgment, self-compassion, and a reduction in mental ruminations [82]. Indeed, recent studies have shown that mindfulness improves emotional stability and mood in healthy subjects [33,82,83,84] and patients with, among the others, treatment-resistant depression [85], chronic pain [86], cancer [87], and spinal cord injury [88]. Also in PD patients, mindfulness-based training was found to reduced depression score [89]. In 2019, Kwok et al., 2019 conducted an RCT on idiopathic PD, comparing levels of anxiety and depression of 71 patients who received mindfulness-based yoga training with 67 patients who received stretching and resistance exercises [90]. The authors showed that patients in the experimental group experienced reduced anxiety and depressive symptoms and increased spiritual well-being and quality of life. Similar effects were reported for MCI patients [91,92]. However, although based on mindfulness practice, our experimental protocol was quite different from the one applied in these studies, being more focused on reminiscence, autobiographical memory, and metacognitive training. This might have affected the results in depression scale. Our exploratory research presents some limits. First, methodological (e.g., limited sample size) and statistical (e.g., absence of corrections during multiple comparison analysis) issues might have affected the results and interpretation. In addition, the new experimental training program is mindfulness-based, but provides a multiapproach treatment (i.e., mindfulness meditation, autobiographical memories recalling and cognitive training). This represents an undeniable clinical advantage, but it might mislead the effect of the treatment. Furthermore, we recruited PD and MCI patients without stratifying or controlling for symptoms, duration, or gravity of pathology. This might have created a heterogenous sample, confounding the results. Lastly, we used only clinical assessments susceptible to assessor biases. As performed in other studies [23,24] structural or functional imaging technique (e.g., MRI or fMRI) evaluation would help defining the neurophysiological substrate of clinical improvement, potentially confirming neuroplastic changes. 

## 5. Conclusions

In conclusion, our innovative experimental multiapproach protocol based on mindfulness training might help enhance residual cognitive functions (i.e., memory, executive functions, attention, concentration, working memory, and language functions) in patients with PD and MCI, compared to a structured program for cognitive rehabilitation. Thus, our protocol might represent a promising rehabilitative intervention to maintain residual cognitive resources, in pathologies that have no current resolutive treatment. Further studies are needed with larger and more homogenous sample sizes, and with optimized study design (double-blinded, parallel design) and outcomes (clinical, neurophysiological, and instrumental assessments).

## Figures and Tables

**Figure 1 brainsci-13-00201-f001:**
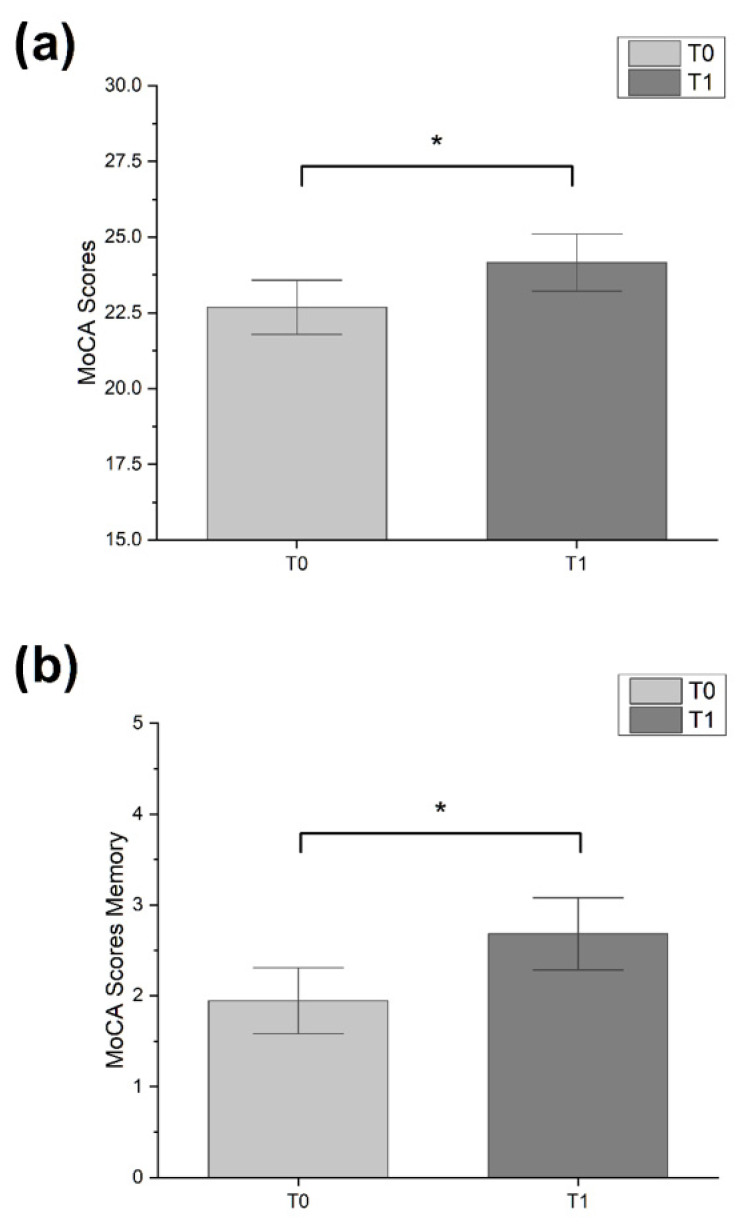
Bar-graphs showing significant changes over time. (**a**) MoCa scores in experimental group. (**b**) MoCa memory sub-scale score in experimental group. Error bars represent standard error (* *p* < 0.05; Friedman test).

**Figure 2 brainsci-13-00201-f002:**
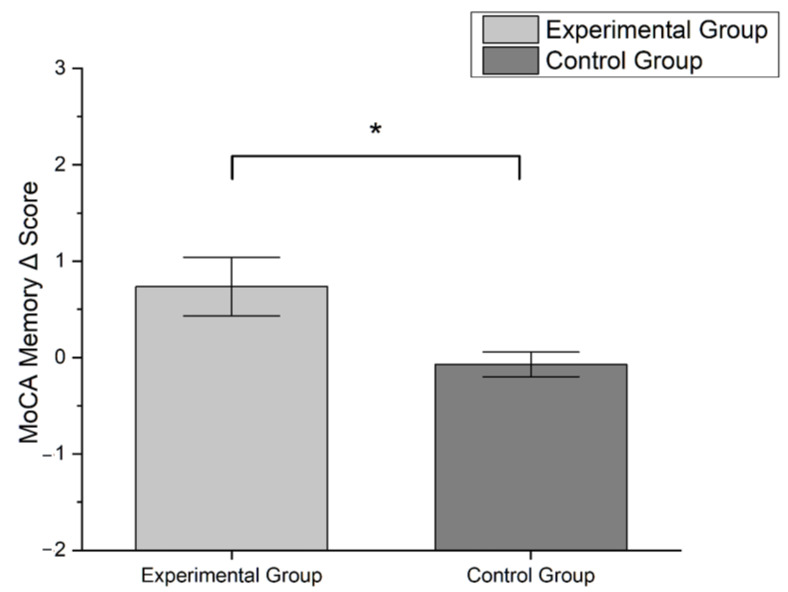
Bar-graph showing differences in Δ MoCA memory sub-scale score between groups according to the formula: Δ score = T1 score − T0 score. Error bars represent standard error (* *p* < 0.05; Mann–Whitney *U* test).

**Table 1 brainsci-13-00201-t001:** Demographic characteristics of sample.

	Total (n = 33)	Control Group (n = 14)	Experimental Group (n = 19)
Females—n (%)	15 (45.4)	6 (42.9)	9 (47.4)
Age—years (mean ± SD)	70.5 ± 11	72.7 ± 11.5	68.8 ± 10.6
Education—years (mean ± SD)	11.58 ± 3.39	10.6 ± 3.2	12.31 ± 3.4
Diagnosis MCI—n (%) PD—n (%)	22 (66.6) 11 (33.3)	8 (57.1) 6 (42.9)	14 (73.6) 5 (26.4)

MCI = mild cognitive impairment; PD = Parkinson’s disease.

**Table 2 brainsci-13-00201-t002:** Neuropsychological assessment results per group over time.

	Experimental Group				Control Group			
	T0	T1	χ2	*p*	T0	T1	χ2	*p*
MoCA	23.00 ± 6.00	26.00 ± 7.00	4.000	0.046 *	20.00 ± 5.75	18.50 ± 7.00	0.091	>0.05
MoCA—memory	2.00 ± 3.00	2.00 ± 2.00	4.571	0.033 *	0.00 ± 1.75	0.00 ± 1.75	0.333	>0.05
MoCA—visuospatial ability	3.00 ± 2.00	3.00 ± 2.00	0.667	>0.05	2.00 ± 1.00	2.50 ± 2.00	0.111	>0.05
MoCA—executive functions	3.00 ± 2.00	3.00 ± 1.00	3.600	>0.05	2.50 ± 1.00	2.00 ± 1.00	0.400	>0.05
MoCA—attention	5.00 ± 1.50	6.00 ± 1.00	0.400	>0.05	5.00 ± 0.75	4.00 ± 1.00	2.667	>0.05
MoCA—language	5.00 ± 2.00	5.00 ± 1.00	1.286	>0.05	4.50 ± 1.75	4.00 ± 1.00	0.143	>0.05
MoCA—orientation	6.00 ± 1.00	6.00 ± 1.00	0.000	>0.05	6.00 ± 1.00	6.00 ± 1.00	0.000	>0.05
BDI-II	17.00 ± 16.50	13.00 ± 13.50	0.000	>0.05	15.00 ± 8.00	15.00 ± 10.75	0.000	>0.05
BDI-II, cognitive and affective factors	9.00 ± 11.50	6.00 ± 9.00	4.000	0.046*	9.00 ± 4.50	8.50 ± 10.75	2.273	>0.05
BDI-II, somato-affective factors	6.00 ± 7.50	6.00 ± 5.00	0.067	>0.05	6.00 ± 2.50	6.00 ± 2.75	0.091	>0.05

Data are expressed as median ± IQR. MoCA = Montreal Cognitive Assessment Test; BDI-II = Beck Depression Inventory-II. The statistical significance refers to the comparison between T0 and T1 (** p < 0.05*; Friedman test).

**Table 3 brainsci-13-00201-t003:** Variations of MoCa and BDI scores in each group, according to the formula Δ score = T1 score − T0 score.

	Experimental Group	Control Group	U	*p*
Δ MoCA	2.00 ± 3.00	0.00 ± 4.50	178.50	>0.05
Δ MoCA—memory	1.00 ± 1.50	0.00 ± 0.00	190.50	0.035 *
Δ MoCA—visuospatial ability	0.00 ± 0.50	0.00 ± 0.75	121.50	>0.05
Δ MoCA—executive functions	0.00 ± 1.00	0.00 ± 1.75	177.00	>0.05
Δ MoCA—attention	0.00 ± 1.00	0.00 ± 1.00	171.50	>0.05
Δ MoCA—language	0.00 ± 0.50	0.00 ± 0.75	141.00	>0.05
Δ MoCA—orientation	0.00 ± 0.00	0.00 ± 0.00	129.00	>0.05
Δ BDI-II	0.00 ± 6.00	0.00 ± 6.50	131.50	>0.05
Δ BDI-II, cognitive and affective factors	−1.00 ± 3.50	−1.00 ± 3.00	124.00	>0.05
Δ BDI-II, somato-affective factors	0.00 ± 4.50	0.00 ± 2.75	137.00	>0.05

Data are expressed as median ± IQR. MoCA = Montreal Cognitive Assessment Test; BDI-II = Beck Depression Inventory-II. The statistical significance refers to the comparison between experimental group and control group (* *p* < 0.05; Mann-Whitney *U* test).

## Data Availability

Data not provided in the article because of space limitations will be shared at the request of other investigators for purposes of replicating procedures and results.

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
