# Peer review of "The Effects of a New Integrated and Multidisciplinary Cognitive Rehabilitation Program Based on Mindfulness and Reminiscence Therapy in Patients with Parkinson’s Disease and Mild Cognitive Impairment: A Pilot Study"

_brainsci, 2023, doi:10.3390/brainsci13020201_

Round 1
Reviewer 1 Report
The authors used randomized, controlled, exploratory pilot study to investigated the the effects of a new integrated and multidisciplinary cognitive rehabilitation program based on mindfulness and reminiscence in patients with Parkinson disease and Mild Cognitive Impairment. They found intitial evidence to support that mindfulness based rehabilitation program could be effective in the treatment of patients affected by Parkinson’s disease and Mild Cognitive Impairment.
Overall, this plot study is very interesting and has implications for clinical context. I only have two minor suggestions.
1. Did the author register this stuty on clinical trial register website?
2. The authors conducted many times of statistical analyses. Multiple comparision should be considered. Based on the p values reported in table 2, the reported effect cannot be survival after correction, such as FDR correction, but it can be understood given the exploratory nature of this study. But should be acknowlegmented in the limitation section.
Author Response
Response to Reviewer 2 Comments
The authors used randomized, controlled, exploratory pilot study to investigated the the effects of a new integrated and multidisciplinary cognitive rehabilitation program based on mindfulness and reminiscence in patients with Parkinson disease and Mild Cognitive Impairment. They found intitial evidence to support that mindfulness-based rehabilitation program could be effective in the treatment of patients affected by Parkinson’s disease and Mild Cognitive Impairment.
Overall, this plot study is very interesting and has implications for clinical context. I only have two minor suggestions.
- Did the author register this stuty on clinical trial register website?
We thank the reviewer for the question. We did not register the study on clinical trial register website, being this is exploratory study and meant as first step for a further, more solid clinical studies.
- The authors conducted many times of statistical analyses. Multiple comparision should be considered. Based on the p values reported in table 2, the reported effect cannot be survival after correction, such as FDR correction, but it can be understood given the exploratory nature of this study. But should be acknowlegmented in the limitation section.
We thank the reviewer for the comment. We added a specific mention in the limitation section (PAGE 8, LINE 240-242)

Reviewer 2 Report
The paper entitled: “The effects of a new integrated and multidisciplinary cognitive rehabilitation program based on mindfulness and reminiscence in patients with Parkinson disease and Mild Cognitive Impairment: a pilot study” is well-written and will be helpful for clinicians.
The authors assessed the efficacy of mindfulness training combined with reminiscence activities for the improvement of cognitive function and behavior of Parkinson disease and Mild Cognitive Impairment patients.
I have some minor comments:
• Authors need to change the visual presentation of Figure 1 to look more proportional.
Line 102 Authors need to substitute the phrase “education years” with “years of education” and explain what level of education they mean (i.e., primary/middle/high school/college, etc.).
Author Response
Response to Reviewer 1 Comments
The paper entitled: “The effects of a new integrated and multidisciplinary cognitive rehabilitation program based on mindfulness and reminiscence in patients with Parkinson disease and Mild Cognitive Impairment: a pilot study” is well-written and will be helpful for clinicians.
The authors assessed the efficacy of mindfulness training combined with reminiscence activities for the improvement of cognitive function and behaviour of Parkinson disease and Mild Cognitive Impairment patients.
I have some minor comments:
- Authors need to change the visual presentation of Figure 1 to look more proportional.
We thank the reviewer for the comment. We decided to use different scales for Y axis of bar-graphs in Figure 1 because bar-graph a) represents the total MoCa score (total points: 30), while bar-graph b) represents the subscale “memory” of MoCa (total point: 5).
- Line 102 Authors need to substitute the phrase “education years” with “years of education” and explain what level of education they mean (i.e., primary/middle/high school/college, etc.).
We thank the reviewer for the comment. We replace the terms “education years” with “years of education” and we explained the level of education (PAGE 3, LINE 102-103).
